# Use of Smart Glasses (Assisted Reality) for Western Australian X-ray Operators’ Continuing Professional Development: A Pilot Study

**DOI:** 10.3390/healthcare12131253

**Published:** 2024-06-24

**Authors:** Curtise K. C. Ng, Moira Baldock, Steven Newman

**Affiliations:** 1Curtin Medical School, Curtin University, GPO Box U1987, Perth, WA 6845, Australia; steven.newman@health.wa.gov.au; 2Curtin Health Innovation Research Institute (CHIRI), Faculty of Health Sciences, Curtin University, GPO Box U1987, Perth, WA 6845, Australia; 3Western Australia Country Health Service, Grace Vaughan House, 233 Stubbs Terrace, Shenton Park, WA 6008, Australia; moira.baldock@health.wa.gov.au; 4South Metropolitan Health Service, 14 Barry Marshall Parade, Murdoch, WA 6150, Australia

**Keywords:** competence, Google Glass, medical imaging, nurse, radiography, radiology, rural health, telemedicine, videoconferencing

## Abstract

Previous studies have explored use of smart glasses in telemedicine, but no study has investigated its use in teleradiography. The purpose of this study was to implement a six-month pilot program for Western Australian X-ray operators (XROs) to use smart glasses to obtain assisted reality support in their radiography practice from their supervising radiographers, and evaluate its effectiveness in terms of XROs’ competence improvement and equipment usability. Pretest–posttest design with evaluation of the XROs’ competence (including their X-ray image quality) and smart glasses usability by XROs in two remote centers and their supervising radiographers from two sites before and after the program using four questionnaire sets and X-ray image quality review was employed in this experimental study. Paired *t*-test was used for comparing mean values of the pre- and post-intervention pairs of 11-point scale questionnaire and image quality review items to determine any XROs’ radiography competence improvements. Content analysis was used to analyze open questions about the equipment usability. Our study’s findings based on 13 participants (11 XROs and 2 supervising radiographers) and 2053 X-ray images show that the assisted reality support helped to improve the XROs’ radiography competence (specifically X-ray image quality), with mean post-intervention competence values of 6.16–7.39 (out of 10) and statistical significances (*p* < 0.001–0.05), and the equipment was considered effective for this purpose but not easy to use.

## 1. Introduction

Australian X-ray operators (XROs) are healthcare workers, often nurses, approved by their jurisdiction regulators, such as Radiological Council of Western Australia (WA), to undertake a limited range of basic radiographic examinations such as chest and extremity X-rays in rural/remote areas where a radiographer is not available. This unavailability of radiographers can be attributed to reasons including the challenge associated with full-time radiographer recruitment, a radiographer’s position being deemed redundant as a result of small rural population, etc. [1]. The XRO model enables patients in rural/remote regions to receive a basic radiography service in their local areas and, hence, avoid long-distance travel and unnecessary financial, emotional and social burdens. However, from the XROs’ perspective, the radiography duty is an extra role beyond their primary profession which requires a transfer of skills. This extended role together with geographical isolation causes various challenges faced by the XROs, including a lack of professional support and continuing professional development (CPD) opportunities, a communication barrier and responsibility overload. These result in potential impacts on their wellbeing, service quality and patient safety [1].

Currently, the major employer of WA XROs, WA Country Health Service (WACHS), arranges radiographers from larger clinical (expert) centers to provide telephone support to their XROs practicing in smaller rural/remote centers to address these issues and as a means for the XROs’ CPD. Although the telephone support is common for traditional telehealth practice [2], our previous qualitative study on WA XROs’ competence, and barriers and facilitators to their radiography practice published in 2020 identified that videoconferencing (VC) support appears to be a better approach [1]. This kind of VC support for XROs was proved effective for improving quality of X-ray images produced by XROs in another Australian state (Queensland) in 2017 [3]. However, the major issues with the traditional VC platform such as the one used in the Queensland’s XROs study [3] are a limited field of view of the fixed camera for the experts (supervising radiographers) to understand clinical situations faced by their XROs and the requirement of hand control for operating the VC equipment [2,4].

Recently, the use of smart glasses in healthcare has become popular despite its first use being reported in 2013 when Google Glass became available [2,4,5]. Smart glasses are a head-mounted computing device with wireless connectivity, a camera, a video display and a headset attached to a frame for a user to wear it like traditional eyeglasses. Unlike the traditional camera, the smart glasses allow the wearer to offer their first-person view (without any blind spots) to remote experts through the head-mounted camera and integrated VC software. The remote experts can provide real-time audio and/or visual guidance on managing various situations (via the same VC platform) received by the wearer through its headset and video display. Its control can be completely handsfree through voice and/or motion recognition [2,4]. Examples of its recent application areas in healthcare include basic life support [6]; medical [7] and nursing student trainings [8,9,10]; emergency medical service delivery [11,12]; telehealth and telemedicine practice for metropolitan, rural and remote areas involving general and specialist physicians such as neuroradiologists, neurosurgeons and pediatric ophthalmologists; nurses; and emergency medical technicians [5,13,14,15,16]. Hence, these uses indicate that the smart glasses could be considered a better technological solution for the XROs in WA rural/remote areas to seek professional support from their supervising radiographers in the larger clinical centers to undertake challenging radiography examinations, and improve their radiographic skills, including X-ray image quality, over time as a better CPD channel when compared with the traditional VC support reported in the aforementioned Queensland XROs’ CPD study [3].

Nevertheless, a systematic review on the use of smart glasses in telemedicine covering 21 studies published in 2023 revealed that there are several barriers for successful implementation of smart glasses in telemedicine, such as ergonomics, human factors, technical limitations, and organizational, security and privacy issues, and no study has investigated the use of smart glasses in teleradiography as yet [2]. The purpose of this study was to implement a pilot program for WA XROs to use smart glasses for obtaining remote support in their radiography practice from their supervising radiographers, and evaluate the effectiveness of this assisted reality support in terms of the XROs’ competence improvement and usability of the assisted reality equipment. It was hypothesized that the assisted reality support helped to improve the WA XROs’ radiography competence and X-ray image quality, and the equipment was considered easy to use and effective.

## 2. Materials and Methods

This study was an experimental study with methods similar to the one by Rawle et al. [3]. Pretest–posttest design with evaluation of the XROs’ competence (including their X-ray image quality) and usability of smart glasses by XROs and supervising radiographers before and after the pilot program was employed. Two WACHS remote clinical centers with relatively large numbers of radiography cases performed by their XROs per year and their (two) corresponding supervising sites (expert centers) were selected for piloting the assisted reality support for six months (1 October 2023–30 March 2024). These centers were chosen because 1. WACHS was the major employer of WA XROs; 2. approximately 1000 cases were performed by the two XROs’ centers per year in total; and 3. all four centers were far (about 1400–2500 km) away from the WA capital, Perth [1]. The required sample size was calculated using Equation (1) [17].
(1)$$n=\frac{2{\left(Z_{\alpha }+Z_{1-\beta }\right)}^{2}\sigma^{2}}{\Delta^{2}}$$
where *n* is the sample size required; *Z*_α_ is 1.96 for a two-tailed test with a significance level of 0.05; *Z*_1−β_ is 0.8416 for a power of 80%; estimated σ is 1.2; and estimated effect size is 20% based on the similar study by Rawle et al. [3].

According to the calculation, 565 X-ray images were required for image quality review per arm, i.e., 1130 images in total. The smart glasses used in our pilot program were RealWear Navigator 500 (Vancouver, Washington, USA) [10,13,18,19]. The study was conducted in accordance with the Declaration of Helsinki, and approved by the WACHS Human Research Ethics Committee (HREC) and Research Governance Unit (project reference number is RGS0000005633 and dates of approvals were 27 October 2022 and 2 December 2022), and Curtin University HREC (approval number is HRE2022-0610 and date of approval was 28 October 2022). Written informed consent was obtained from all participants including XROs, radiographers and patients involved in the study except for the retrospective review of patients’ X-ray images taken before the intervention with a wavier of consent approved by the WACHS HREC and Research Governance Unit, and Curtin University HREC [3].

### 2.1. Participant Selection

All XROs, and their patients and supervising radiographers, of the four centers were invited to participate in this pilot program. The following were the participants’ inclusion and exclusion criteria [3].

#### 2.1.1. Inclusion Criteria

##### XROs

-Approved by the Radiological Council of WA as XROs;-Employed by WACHS.

##### Supervising Radiographers

-Registered with Australian Health Practitioner Regulation Agency as diagnostic radiographers;-Appointed by the Radiological Council of WA as supervising approved radiographers;-Employed by WACHS.

##### Patients

-Pediatric and adult patients with X-ray examinations performed by the XROs between 1 April 2023 and 30 September 2023 (pre-intervention period for the retrospective X-ray image quality review), and between 1 October 2023 and 30 March 2024 (intervention period).

#### 2.1.2. Exclusion Criteria

##### XROs, Supervising Radiographers and Patients

-Refusal to consent to participation/unable to obtain consent.

### 2.2. Assisted Reality Support Program

A train-the-trainer model was adopted for the assisted reality equipment vendor to train our research team, and subsequently, our team trained the XROs and their supervising radiographers to appropriately use the equipment before implementing the assisted reality support in the clinical practice [10,20]. Each training session took place at either the participants’ centers or our university with a duration of 1–2 h. Figure 1 shows one of our equipment training sessions with an XRO to wear the RealWear Navigator 500 smart glasses to perform a simulated hand X-ray examination under the guidance of her supervising radiographer. During that training session, both a smartphone (Galaxy S22, Samsung Group, Yeongtong-gu, Suwon, South Korea) and an Apple iPad 10.2” (9th generation) with 64-gigabyte data storage and WiFi plus cellular connectivity (Cupertino, CA, USA) were used by the supervising radiographer to provide the guidance through Microsoft Teams (version 4.20.1, Microsoft Corporation, Redmond, WA, USA), which was the only VC platform approved by WACHS for clinical use. However, for the assisted reality support intervention, only the Apple iPads were employed for the supervising radiographers to use Microsoft Teams to provide the guidance [21]. All XROs and their supervising radiographers were provided one-page quick-reference guides for requesting and providing the assisted reality support after the training, respectively [19].

Each supervising radiographer’s site was given one iPad (Apple Inc., Cupertino, CA, USA) and every XRO’s center was provided one smart glasses device (RealWear Inc., Vancouver, WA, USA). Both iPad and smart glasses were connected to their centers’ network through WiFi with a fourth-generation broadband cellular network (4G) as a backup internet connection [2]. The XROs only used the assisted reality support when they encountered challenging cases for performing radiographic examinations and/or image quality reviews during the intervention, aligning with the existing WACHS protocol for the telephone support [1].

### 2.3. Evaluation of Assisted Reality Support Program

Four sets of questionnaires (pre- and post-intervention questionnaires for the XROs and supervising radiographers) were developed based on Andersson et al.’s [22,23] validated radiographers’ competence scale (designed for radiographers with dual registrations as radiographers and nurses [22,23,24]) with additional open questions about usability of the assisted reality equipment (based on Yoon et al.’s [10] study on using assisted reality to provide remote support for nursing student training). Andersson et al.’s [22,23] and Yoon et al.’s [10] questionnaires were selected for developing our evaluation questionnaires because they were designed for nurses and, hence, matched our XROs’ characteristics which were predominantly nurses. The questionnaire was piloted with three XROs and two radiographers not directly involved in our study, resulting in several revisions for improving its reliability and validity [25]. The developed questionnaires were delivered to the XROs and supervising radiographers to assess the XROs’ radiography competence (using an 11-point scale, 0—no competence; 1—extremely low competence; 2—very low competence; 3—low competence; 4—just below minimally acceptable competence; 5—minimally acceptable competence; 6—just above minimally acceptable competence; 7—competence; 8—high competence; 9—very high competence; 10—outstanding competence) before and after the intervention [22,23,24]. Rawle et al.’s [3] 11-point image quality grading scale (0—no attempt to meet quality requirement;1—extremely poor quality; 2—very poor quality; 3—poor quality; 4—just below minimally acceptable diagnostic quality; 5—meeting minimally acceptable diagnostic quality; 6—just above minimally acceptable diagnostic quality; 7—good quality; 8—very good quality; 9—excellent quality; 10—outstanding quality) was used to assess the quality of X-ray images acquired by the XROs prior to and during the intervention through a picture archiving and communication system (PACS) [25,26]. Only one observer (who was a current WACHS-approved radiographer and a former area chief medical imaging technologist (MIT)) was involved in this image quality review process to avoid inter-observer variability [3]. The image quality grading scale was piloted with two radiographers not involved in this study to improve its reliability and validity before administration [25].

### 2.4. Data Analysis

SPSS Statistics 29 (International Business Machines Corporation, Armonk, NY, USA) was used for statistical analysis. For multiple choice items (demographics questions of the questionnaires and X-ray image information questions of the image quality assessment forms), percentage of frequency was used for data analysis. For the 11-point scale items of the four questionnaires and two image quality assessment forms, mean and standard deviation were calculated and a paired *t*-test was used for comparing the mean values of the pre- and post-intervention pairs of the 11-point scale items to determine any XROs’ radiography competence (including image quality) improvements and enable findings comparison with the similar study [3,27,28]. A *p*-value less than 0.05 represented statistical significance [3,26,29,30,31]. Content analysis with quasi-statistics as an accounting system was used to analyze the open questions about the usability of the assisted reality equipment [25,32].

## 3. Results

All XROs (total: 11; site 1: 6; site 2: 5) and supervising radiographers (total: 2; 1 per site) of the four selected centers were recruited and completed the assisted reality support program training and pre-intervention questionnaires afterwards, yielding 100% response rates. One XRO withdrew shortly after the training due to her prescription glasses being incompatible with the smart glasses. During the intervention, the numbers of XROs of the two centers fluctuated, consistent with the usual WACHS staffing arrangement. At the end of the intervention, six XROs (3 per site) were rostered to perform radiography duty. Half of the rostered XROs and all supervising radiographers returned the post-intervention questionnaires, resulting in 50% and 100% response rates, respectively. Table 1 shows their demographics.

Table 2 and Table 3 show the participants’ perceptions of XRO radiographic competences with no statistically significant difference before and after the assisted reality support program (*p* = 0.099–1.000). However, notable mean value decreases are noted for the “participating in quality improvement for patient safety and care” and “adapting examination based on patient’s needs” competences under the patient care (Table 2) and technical processes categories (Table 3) after the intervention, respectively. Moreover, these two competences and the other five (more than half) technical process-related competence (“adapting examination based on patient’s needs”, “reducing radiation doses for patients and staff”, “producing accurate and correct images”, “evaluating image quality against referral and clinical question” and “optimizing image quality”) levels of XROs were perceived minimally acceptable (level 5) or just above this (level 6) overall after the program.

Table 4 demonstrates the participants’ perceptions of assisted reality support and equipment usability before and after the intervention. The technical performances of the equipment were well perceived after the program. These included the audio and video quality and data transmission speed enabling the supervising radiographers to obtain adequate ideas about the XROs’ situations. However, 60% of participants preferred the telephone (smartphone) support to the assisted reality support as a result of 100% of participants indicating issues of ill-fitting headband, voice control, login and non-intuitive design of smart glasses after the intervention.

Table 5 illustrates the types, projections, patient ages, image receptor (computed radiography cassette) sizes and exposure factors (kV and milliampere-seconds) of the X-ray examinations performed by the two centers’ XROs within six months before (the numbers of examinations and images were 487 and 982, respectively) and during the six-month intervention period (the numbers of examinations and images were 495 and 1071, respectively). Their frequencies and proportions before and during the program were comparable. Table 6 shows the quality of X-ray images before and during the intervention. Statistically significant improvements of X-ray image quality are noted for all criteria with about half (inclusion of required anatomy, side marker and image quality regarding artifact) determined good quality and the others meeting just above minimally acceptable diagnostic quality requirements during the program (*p* < 0.001–0.05). In contrast, about half of the areas (beam collimation, image quality regarding exposure and overall diagnostic value for pathology identification) only met the minimally acceptable diagnostic quality requirements before the intervention.

## 4. Discussion

Over the last few years, numerous studies have investigated the use of smart glasses (with [33,34,35,36,37,38,39,40,41,42,43,44,45,46] and without augmented reality [5,6,7,8,9,10,11,12,13,14,15,16]) in healthcare. Given that our study participants lived in rural/remote areas with an expected digital divide issue [47,48,49], the more advanced use of smart glasses, i.e., augmented reality with superimposing virtual objects on video of real world was not used for our XROs to obtain remote support in their radiography practice from their supervising radiographers [33,34,35,36,37,38,39,40,41,42,43,44,45,46]. Among the recent studies on the use of smart glasses without any augmented reality [5,6,7,8,9,10,12,13,14,15,16], only four evaluated its use with real patients and their sample sizes were 8 [15], 37 [16], 103 [5] and 622 [14], respectively. For the other studies [6,7,8,9,10,12,13], only one to six simulated clinical scenarios were involved. In contrast, our study covered 497 patients during the intervention period, which could be considered a strength. Also, according to the systematic review on the use of smart glasses in telemedicine with the inclusion of 21 studies published in 2023 [2], our study is the first one to investigate the use of smart glasses in teleradiography.

Our study findings presented in Table 6 demonstrate that the use of smart glasses helped our study’s XROs to significantly improve their radiography competence and X-ray image quality with at least just above minimally acceptable diagnostic quality rating (six) for all aspects during the program (*p* < 0.001–0.05). The average increase in these mean values of image quality scores was 0.59. Our findings match the ones of the Queensland’s study on using traditional VC platform for teleradiography and an 11-point scale to evaluate the quality of 326 pre-intervention and 234 intervention X-ray images acquired by their XROs with statistically significant improvements for all image quality criteria, with all but one at least rating six, and average increase in mean values being 0.6. It is noted that their “appropriate collimation” criterion only had a mean rating of 5.9 during their intervention [3].

Nonetheless, Table 2 and Table 3 reveal that our assisted reality support program did not support the XROs in improving their radiography competence. More concerningly, mean value decreases are noted in many aspects of their radiography competence after the intervention although there was no statistically significant difference for all competence items. It is well known that the use of questionnaire for competence assessment might not be reliable. For example, Graves et al. [50] asked 140 medical students to indicate their self-perceived competence before and after their objective structured clinical examination (OSCE) through the use of questionnaires. They found that there was a decrease in their students’ self-perceived competence after the OSCE and their students’ competence ratings were weakly correlated with the corresponding OSCE results. Hence, our findings shown in Table 2 and Table 3 should be used with caution.

As per the aforementioned systematic review on the use of smart glasses in telemedicine [2], there are four major challenges, ergonomics and human factors, technical limitations, organizational factors, and security and privacy issues affecting the usability of smart glasses in clinical environment. Their details are as follows.


Ergonomics and Human Factors


Prescription glasses incompatible with smart glasses;Smart glasses camera range and gaze direction misaligned;Voice control issue;Smart glasses as distraction.


Technical Limitations


Network stability and bandwidth issues;Low battery capacity;Small video display size;Background noise not removed;Ambient lighting affecting video quality of display and camera;Program (including video streaming) interface not user friendly.


Organizational Factors


Extra workload;Expensive equipment;Extensive equipment training required.


Security and Privacy Issues


Data breach;Patient privacy violation.

Our study’s results show that the above issues, except most of the items under the categories of technical limitations (network stability and bandwidth issues, low battery capacity, small video display size and background noise not removed) and security and privacy issues (patient privacy violation and data breach), were also reported by our participants (Table 4). For our intervention, 4G internet connection and extra batteries were arranged for the XROs as backup. Moreover, our participants indicated that the smart glasses video quality was sufficient. Although the RealWear smart glasses were able to remove the background noise, one of our supervising radiographers expressed his need of hearing the patients for the assisted reality support. No patient privacy violation and data breach occurred in our intervention because a written informed consent was obtained from each patient after explaining to them that the video was only viewed (but not recorded) by the supervising radiographers, consistent with the standard-of-care procedure. Moreover, two-factor authentication (2FA) was required to use Microsoft Teams on the smart glasses and iPads. However, these measures for addressing the patient privacy and data security requirements together with the issues of voice control and non-intuitive software interface design of the smart glasses significantly increased the workload of XROs who considered this as a distraction (interfering with radiographic examination process). This is because extra time and effort were required to obtain consent, complete 2FA and use the smart glasses via voice control with unfamiliar software interface [2,51]. These factors could explain why one XRO and both supervising radiographers preferred the smart phones to the current assisted reality equipment.

Apparently, the design of smart glasses needs to be further improved to become more intuitive for promoting its use in healthcare. Recent research has already started exploring this. For example, Zhang et al. [11] conducted a study in 2022 to determine the implications of smart glasses design for its wider adoption in emergency medical services. However, the ill-fitting headband and voice control issues can be readily addressed to a certain extent by using other head mounts for smart glasses, such as a cap [52], and a wireless keyboard [53], despite voice control being one of the benefits (reasons) of its use in a healthcare environment. Moreover, when funding is available, a smart glasses device can be assigned to each XRO. In this way, they can connect their assigned devices to Microsoft Teams at the beginning of their shifts to avoid the 2FA login issue occurring during radiographic examinations [51]. Also, smart phones with high mobility can be given to supervising radiographers to provide the assisted reality support anywhere and anytime [54]. In addition, it would be interesting to match the use of smart glasses with smartphone apps specifically tailored to various fields of medicine to understand their potential mutual application in the future [55,56].

This study had three major limitations. Only three XROs and two supervising radiographers completed the post-intervention questionnaires. However, 982 patients’ X-ray examination data were used for evaluating the XROs’ competence development, which was greater than the similar study on using the traditional VC platform for XROs’ CPD in Queensland [3]. Also, for many studies about using the smart glasses in healthcare, only one to six simulated clinical scenarios instead of real patients were employed for the evaluation [6,7,8,9,10,12,13]. Moreover, our non-patient participant number exceeded some of the similar smart glasses studies’ ones which were as low as two participants. This resulted in our study’s smart glasses usability evaluation findings matching those reported in the systematic review on the use of smart glasses in telemedicine [2]. Although our intervention period was in line with the Queensland XROs’ study, which was six months, and Table 5 shows the frequencies and proportions of examination types, projections and patient ages before and during our program were comparable, it would be better if the intervention period was longer for implementing the aforementioned remedies, such as providing other smart glasses head mount options and wireless keyboards to XROs and smart phones to supervising radiographers, and evaluating their effectiveness. Besides, as per the human research ethics requirements, patients who were unable to provide consent during the intervention were excluded from our study, but this situation was the same as the one of Rawle et al.’s study [3].

## 5. Conclusions

Our study’s findings show that the assisted reality support helped to significantly improve the WA XROs’ radiography competence (specifically X-ray image quality) and the equipment was considered effective for this purpose. Nonetheless, our participants indicated that the equipment was not easy to use due to the ill-fitting headband, voice control and login issues of the smart glasses, and lower mobility of iPads. The remedies including use of other head mounts, wireless keyboards and smart phones, and completion of 2FA at the beginning of shifts rather than radiographic examinations are recommended to potentially address these issues. Further research is needed to evaluate the effectiveness of these recommendations. Also, future studies should be conducted to improve the design of smart glasses to become more intuitive and user friendly for promoting its use in healthcare.

## Figures and Tables

**Figure 1 healthcare-12-01253-f001:**
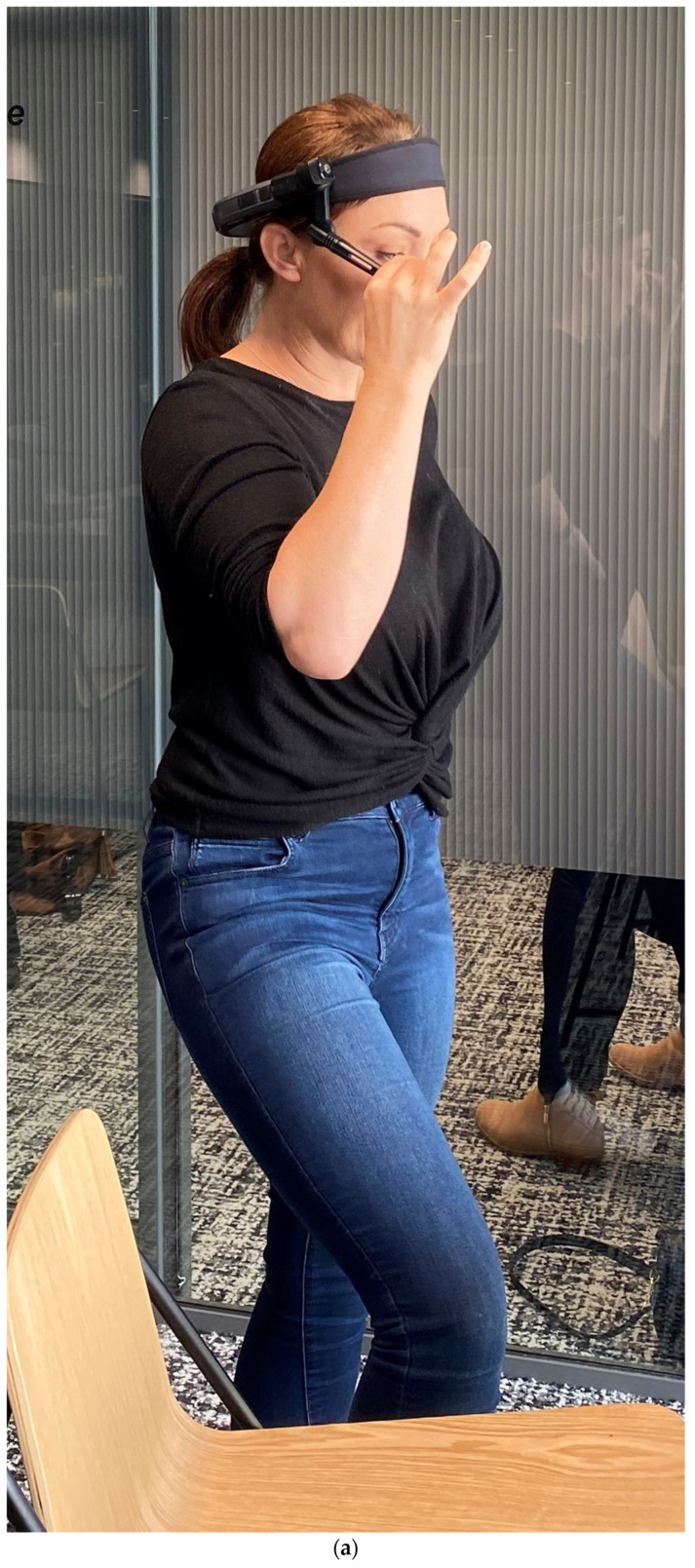
Equipment training session for (**a**) X-ray operators to complete a simulated hand X-ray examination with assisted reality support from (**b**) their supervising radiographer at our university.

**Table 1 healthcare-12-01253-t001:** Summary of Demographic Information of X-ray Operators (*n* = 11) and Supervising Radiographers (*n* = 2).

	Cohort Frequency (%)					
X-ray Operators						
**Primary Profession Position**	Enrolled nurse 0 (0.0%)	Registered nurse 7 (63.6%)	Registered paramedic 2 (18.2%)	Registered GP 0 (0.0%)	Management 2 (18.2%)	Other 0 (0.0%)
**Primary Profession Experience**	0–5 years 1 (9.1%)	>5–15 years 3 (27.3%)	>15–25 years 5 (45.5%)		>25 years 2 (18.2%)	
**Qualification for Primary Profession**	Master’s degree 1 (9.1%)	Bachelor’s degree 8 (72.7%)	Sub-degree 2 (18.2%)		Other 0 (0.0%)	
**Primary Profession Qualification Issuing Country**	Australia 9 (81.8%)		New Zealand 1 (9.1%)		Overseas 1 (9.1%)	
**Training Provider for X-ray Operator Qualification**	Curtin University 10 (90.9%)	WACHS 1 (9.1%)	Radiological Council of WA 0 (0.0%)		Other 0 (0.0%)	
**X-ray Operator Training Completion Year**	1973–2007 0 (0.0%)		2008–2012 0 (0.0%)		>2012 11 (100.0%)	
**Radiography Practice Experience**	0–1 years 5 (45.5%)	>1–2 years 3 (27.3%)	>2–5 years 1 (9.1%)	>5–15 years 2 (18.2%)	>15–25 years 0 (0.0%)	>25 years 0 (0.0%)
**Average Number of X-ray Examinations Performed per Month within the Past Year**	<1 3 (27.3%)	1–5 2 (18.2%)	6–10 2 (18.2%)	11–15 2 (18.2%)	16–20 0 (0.0%)	>20 2 (18.2%)
**Radiographers**						
**Professional Experience**	0–1 years 0 (0.0%)	>1–2 years 0 (0.0%)	>2–5 years 0 (0.0%)	>5–15 years 1 (50.0%)	>15–25 years 1 (50.0%)	>25 years 0 (0.0%)
**Professional Qualification**	Master’s degree 0 (0.0%)		Bachelor’s degree 2 (100.0%)	Sub-degree 0 (0.0%)	Other 0 (0.0%)	
**Course Provider for Professional Qualification**	Curtin University 2 (100.0%)		Other Australian Provider 0 (0.0%)		Overseas Provider 0 (0.0%)	
**Professional Position**	MIT 0 (0.0%)	Senior MIT 2 (100.0%)	Deputy Chief MIT 0 (0.0%)	Chief MIT 0 (0.0%)	Other 0 (0.0%)	

GP, general practitioner; MIT, medical imaging technologist; WA, Western Australia; WACHS, Western Australia Country Health Service.

**Table 2 healthcare-12-01253-t002:** Participants’ perceptions of X-ray operator radiographic competences (related to patient care) before and after assisted reality support program.

Competence Statement	X-ray Operator			Supervising Radiographer			Overall		
	Before (*n* = 11)	After (*n* = 3)	*p*-Value	Before (*n* = 2)	After (*n* = 2)	*p*-Value	Before (*n* = 13)	After (*n* = 5)	*p*-Value
**Carrying out doctor’s prescriptions**	8.27 ± 1.68	9.00 ± 1.73	0.157	8.00 ± 1.41	7.50 ± 0.71	0.795	8.23 ± 1.59	8.40 ± 1.52	0.486
**Applying ethical guidelines**	9.00 ± 1.10	9.00 ± 1.73	0.423	9.00 ^1^	8.50 ± 2.12	- ^1^	9.00 ± 1.04	8.80 ± 1.64	1.000
**Adequately informing patient**	8.91 ± 1.14	9.00 ± 1.73	0.840	8.50 ± 0.71	7.00 ± 0.00	0.205	8.85 ± 1.07	8.20 ± 1.64	0.351
**Guiding and educating patient**	8.55 ± 1.21	8.33 ± 1.53	1.000	7.00 ^1^	7.00 ± 0.00	- ^1^	8.42 ± 1.24	7.80 ± 1.30	0.294
**Empowering patient through having patient involved in examination**	8.64 ± 1.21	8.33 ± 1.53	0.742	8.00 ^1^	7.00 ± 0.00	- ^1^	8.58 ± 1.16	7.80 ± 1.30	0.230
**Guiding patient’s relatives**	8.55 ± 1.21	8.33 ± 1.53	1.000	- ^1^	7.00 ^1^	- ^1^	8.55 ± 1.21	8.00 ± 1.41	0.444
**Encouraging and supporting patient**	8.64 ± 1.21	8.33 ± 1.53	0.742	8.00 ± 0.00	7.00 ± 0.00	- ^1^	8.54 ± 1.13	7.80 ± 1.30	0.230
**Protecting patient’s integrity**	9.00 ± 1.10	9.00 ± 1.73	0.840	8.00 ± 0.00	7.00 ^1^	- ^1^	8.85 ± 1.07	8.50 ± 1.73	0.731
**Alleviating patient’s anxiety**	8.82 ± 1.08	8.67 ± 1.53	1.000	8.00 ± 0.00	7.00 ^1^	- ^1^	8.69 ± 1.03	8.25 ± 1.50	0.608
**Assessing risk associated with leaving patient unattended**	8.64 ± 1.29	8.67 ± 1.53	0.635	9.00 ^1^	7.00 ^1^	- ^1^	8.67 ± 1.23	8.25 ± 1.50	0.854
**Observing and monitoring patient**	9.09 ± 1.04	8.67 ± 1.53	1.000	9.00 ± 0.00	7.00 ^1^	- ^1^	9.08 ± 0.95	8.25 ± 1.50	0.608
**Recognizing patient in shock state**	8.91 ± 0.94	9.00 ± 1.73	0.742	9.00 ± 0.00	7.00 ^1^	- ^1^	8.92 ± 0.86	8.50 ± 1.73	0.391
**Recognizing pain and pain reactions**	9.00 ± 0.77	9.00 ± 1.73	0.742	9.00 ± 0.00	7.50 ± 0.71	0.205	9.00 ± 0.71	8.40 ± 1.52	0.374
**Collaborating with internal and external colleagues**	8.91 ± 0.70	8.33 ± 1.53	0.667	9.00 ^1^	7.00 ± 1.41	- ^1^	8.92 ± 0.67	7.80 ± 1.48	0.242
**Collaborating with other internal and external healthcare professionals**	8.91 ± 0.70	8.00 ± 1.00	0.184	9.00 ^1^	7.50 ± 2.12	- ^1^	8.92 ± 0.67	7.80 ± 1.30	0.242
**Reporting to internal and external colleagues and other healthcare professionals**	8.73 ± 0.90	8.00 ± 1.00	0.225	9.00 ^1^	7.00 ± 1.41	- ^1^	8.75 ± 0.87	7.60 ± 1.14	0.108
**Participating in quality improvement for patient safety and care**	8.36 ± 1.21	5.33 ± 1.53	0.184	7.00 ^1^	7.50 ± 0.71	- ^1^	8.25 ± 1.22	6.20 ± 1.64	0.099

Figures (except *p*-Values) are expressed in mean ± standard deviation (0—no competence; 1—extremely low competence; 2—very low competence; 3—low competence; 4—just below minimally acceptable competence; 5—minimally acceptable competence; 6—just above minimally acceptable competence; 7—competence; 8—high competence; 9—very high competence; 10—outstanding competence). ^1^ Not available due to insufficient data or standard error of difference being zero.

**Table 3 healthcare-12-01253-t003:** Participants’ perceptions of X-ray operator radiographic competences (related to technical processes) before and after assisted reality support program.

Competence Statement	X-ray Operator			Supervising Radiographer			Overall		
	Before (*n* = 11)	After (*n* = 3)	*p*-Value	Before (*n* = 2)	After (*n* = 2)	*p*-Value	Before (*n* = 13)	After (*n* = 5)	*p*-Value
**Organizing and planning examination with consideration of clinical situation**	8.00 ± 1.41	7.00 ± 1.00	0.500	7.50 ± 0.71	7.50 ± 0.71	1.000	7.92 ± 1.32	7.20 ± 0.84	0.338
**Preparing radiographic equipment**	8.00 ± 1.48	7.67 ± 1.15	1.000	8.00 ± 0.00	7.00 ± 0.00	- ^1^	8.00 ± 1.35	7.40 ± 0.89	0.294
**Independently planning and preparing work based on existing documentation**	8.09 ± 1.51	7.67 ± 1.15	1.000	8.00 ^1^	7.50 ± 0.71	- ^1^	8.08 ± 1.44	7.60 ± 0.89	0.621
**Prioritizing patients in workflow**	8.27 ± 1.35	7.67 ± 1.15	0.500	8.00 ± 0.00	7.50 ± 0.71	0.500	8.23 ± 1.24	7.60 ± 0.89	0.242
**Adapting examination based on patient’s needs**	8.18 ± 1.40	6.33 ± 0.58	0.344	8.00 ± 1.41	8.00 ^1^	- ^1^	8.15 ± 1.34	6.75 ± 0.96	0.116
**Performing positioning procedures for more challenging radiographic projections**	6.09 ± 2.12	5.33 ± 1.53	0.500	7.00 ^1^	7.00 ± 0.00	- ^1^	6.17 ± 2.04	6.00 ± 1.41	0.501
**Reducing radiation doses for patients and staff**	7.18 ± 2.09	6.67 ± 1.53	0.500	7.50 ± 0.71	7.00 ± 0.00	0.500	7.23 ± 1.92	6.80 ± 1.10	0.405
**Producing accurate and correct images**	7.00 ± 1.73	7.00 ± 1.00	0.500	7.50 ± 0.71	6.50 ± 0.71	0.500	7.08 ± 1.61	6.80 ± 0.84	0.621
**Evaluating image quality against referral and clinical question**	7.00 ± 1.61	6.67 ± 0.58	1.000	7.50 ± 0.71	6.50 ± 0.71	0.500	7.08 ± 1.50	6.60 ± 0.55	0.468
**Optimizing image quality**	6.55 ± 1.81	6.67 ± 1.53	- ^1^	7.00 ± 1.41	6.50 ± 0.71	0.795	6.62 ± 1.71	6.60 ± 1.14	0.648

Figures (except *p*-Values) are expressed in mean ± standard deviation (0—no competence; 1—extremely low competence; 2—very low competence; 3—low competence; 4—just below minimally acceptable competence; 5—minimally acceptable competence; 6—just above minimally acceptable competence; 7—competence; 8—high competence; 9—very high competence; 10—outstanding competence). ^1^ Not available due to insufficient data or standard error of difference being zero.

**Table 4 healthcare-12-01253-t004:** Participants’ perceptions of assisted reality support and equipment usability before and after intervention.

Question	X-ray Operator (XRO)		Supervising Radiographer	
	Before (*n* = 11)	After (*n* = 3)	Before (*n* = 2)	After (*n* = 2)
Reason of using assisted reality support by XROs	Not applicable	Receiving support for managing complex radiographic examinations: 3 (100.0%)	Not applicable	Providing support for patient positioning, image (receptor) management and interpreting examination request: 2 (100.0%)
Aspect of support received/provided for performing radiographic examination	Not applicable	All aspects: 1 (50.0%) Patient positioning: 1 (50.0%) ^1^	Not applicable	Providing support for patient positioning, image (receptor) management and interpreting examination request: 2 (100.0%)
Easy to use assisted reality equipment	Yes: 7 (63.6%) Unsure: 3 (27.3%) No: 1 (9.1%)	No due to time-consuming login process (including voice control issue): 3 (100.0%)	Yes: 2 (100.0%)	No due to ill-fitting headband, issues of voice control, login and non-intuitive design: 2 (100.0%)
Assisted reality equipment always reliable and available	Yes: 2 (20.0%) Unsure: 7 (70.0%) No: 1 (10.0%) ^1^	No due to login issue: 2 (66.7%) Unsure: 1 (33.3%)	No when compared to smartphone: 2 (100.0%)	No due to unfit headband, accent affecting voice control and non-intuitive design causing technical issues: 2 (100.0%)
Able to set up assisted reality equipment quickly	Yes: 4 (40.0%) Unsure: 4 (40.0%) No: 2 (20.0%) ^1^	No due to login issue (including voice control issue): 3 (100.0%)	Yes: 2 (100.0%)	No due to unfit headband, accent affecting voice control and non-intuitive design causing technical issues: 2 (100.0%)
Assisted reality equipment not interfering radiographic examination process	Yes: 4 (40.0%) Unsure: 3 (30.0%) No: 3 (30.0%) ^1^	No due to long set up time: 3 (100.0%)	Not applicable	Not applicable
Assisted reality equipment providing adequate ideas about clinical situation faced by XROs	Not applicable	Not applicable	Yes: 2 (100.0%)	Yes: 2 (100.0%)
Assisted reality equipment providing adequate ideas about condition of patients managed by XROs	Not applicable	Not applicable	Yes: 2 (100.0%)	Yes: 2 (100.0%)
Assisted reality equipment providing adequate ideas about XROs’ hand movements	Not applicable	Not applicable	Yes: 1 (50.0%) Unsure: 1 (50.0%)	Yes: 2 (100.0%)
Assisted reality equipment providing adequate ideas about radiographic procedures carried out by XROs	Not applicable	Not applicable	Yes: 1 (50.0%) Unsure: 1 (50.0%)	Yes: 2 (100.0%)
Smart glasses video quality sufficient for assisted reality support	Yes: 7 (70.0%) Unsure: 3 (30.0%) ^1^	Yes: 3 (100.0%)	Yes: 2 (100.0%)	Yes when lighting adequate and camera not too close to objects with minimal movement: 2 (100.0%)
Smart glasses audio quality sufficient for assisted reality support	Yes: 5 (50.0%) Unsure: 5 (50.0%) ^1^	Yes: 3 (100.0%)	Yes: 2 (100.0%)	Yes: 1 (50.0%) No as only able to hear XROs but not patients: 1 (50.0%)
Data transmission speed for assisted reality support sufficient for avoiding any lag	Yes: 3 (30.0%) Unsure: 5 (50.0%) No: 2 (20.0%) ^1^	Yes: 3 (100.0%)	Yes: 1 (50.0%) Unsure: 1 (50.0%)	Yes: 2 (100.0%)
Preferring assisted reality support to telephone support	Yes: 2 (20.0%) Unsure: 7 (70.0%) No: 1 (10.0%) ^1^	Yes: 2 (66.7%) No as smartphone requiring less set up time: 1 (33.3%)	Yes: 1 (50.0%) No when compared to smartphone allowing quicker set up: 1 (50.0%)	No as smartphone allowing quicker set up and being more user-friendly and convenient: 2 (100.0%)
Any other comments about assisted reality support	Great for emergency telehealth service: 1 (10.0%) No: 9 (90.0%) ^1^	No: 3 (100.0%)	Challenging to set up initially: 1 (50.0%) No: 1 (50.0%)	No: 2 (100.0%)

^1^ Not answered by one participant.

**Table 5 healthcare-12-01253-t005:** Information of X-ray images acquired within six months before (pre-intervention: *n* = 982) and during assisted reality support program (intervention: *n* = 1071).

	*Cohort* Pre-Intervention Frequency (%)/Intervention Frequency (%)																						
**Examination**	*Chest* 227 (23.1%)/212 (19.8%)	*Upper Extremity* 468 (47.7%)/498 (46.5%)													*Lower Extremity* 287 (29.2%)/359 (33.5%)								*Other* 0 (0.0%)/2 (0.2%)
		*Clavicle* 6 (0.6%)/9 (0.8%)	*Elbow* 32 (3.3%)/44 (4.1%)		*Finger* 44 (4.5%)/49 (4.6%)		*Forearm* 57 (5.8%)/31 (2.9%)		*Hand* 207 (21.1%)/216 (20.2%)	*Humerus* 6 (0.6%)/5 (0.5%)		*Shoulder* 32 (3.3%)/30 (2.8%)	*Thumb* 3 (0.3%)/13 (1.2%)	*Wrist* 81 (8.2%)/101 (9.4%)	*Ankle* 52 (5.3%)/107 (10.0%)		*Calcaneum* 2 (0.2%)/9 (0.8%)	*Foot* 132 (13.4%)/135 (12.6%)	*Knee* 56 (5.7%)/73 (6.8%)		*Lower Leg* 30 (3.1%)/11 (1.0%)	*Toes* 15 (1.5%)/24 (2.2%)	
**Projection**	*AP* 243 (24.7%)/263 (24.6%)					*Lateral* 254 (25.9%)/308 (28.8%)					*Oblique* 159 (16.2%)/178 (16.6%)					*PA* 322 (32.8%)/322 (30.1%)				*Other* 4 (0.4%)/0 (0.0%)			
**Patient Age**	*0–11 months* 1 (0.1%)/3 (0.3%)												*>11 months* 981 (99.9%)/1068 (99.7%)										
													*1–17 years* 165 (16.8%)/120 (11.2%)						*≥18 years* 816 (83.1%)/948 (88.5%)				
**CR Cassette Size**	*Not applicable* 981 (99.9%)/1071 (100.0%)				*18X24 cm* 1 (0.1%)/0 (0.0%)							*24X30 cm* 0 (0.0%)/0 (0.0%)					*35X43 cm* 0 (0.0%)/0 (0.0%)				*Other* 0 (0.0%)/0 (0.0%)		
**kV ^1^**	*<50* 215 (26.7%)/162 (16.6%)			*50–59* 273 (34.0%)/450 (46.1%)				*60–69* 62 (7.7%)/118 (12.1%)				*70–79* 47 (5.8%)/42 (4.3%)		*80–89* 46 (5.7%)/34 (3.5%)				*90–99* 159 (19.8%)/138 (14.1%)			*≥100* 2 (0.2%)/32 (3.3%)		
**mAs ^1^**	*0.5* 1 (0.1%)/2 (0.2%)			*1* 437 (54.4%)/371 (38.0%)				*2* 336 (41.8%)/426 (43.6%)				*3* 0 (0.0%)/140 (14.3%)		*4* 30 (3.7%)/7 (0.7%)				*5* 0 (0.0%)/19 (1.9%)			*6* 0 (0.0%)/11 (1.1%)		

^1^ kV and milliampere-seconds (mAs) information was only available for 804 images. AP, anteroposterior; CR, computed radiography; PA, posteroanterior.

**Table 6 healthcare-12-01253-t006:** Quality of X-ray images acquired within six months before and during assisted reality support program.

Criterion	Image Quality Score		*p*-Value
	Before (*n* = 982)	During (*n* = 1071)	
**Inclusion of Required Anatomy**	6.68 ± 1.90	7.39 ± 1.86	<0.001
**Beam Collimation**	5.88 ± 1.73	6.75 ± 1.43	<0.001
**Side Marker**	6.56 ± 1.95	7.07 ± 1.46	<0.001
**Image Quality Regarding Exposure**	5.65 ± 1.30	6.16 ± 1.22	<0.001
**Image Quality Regarding Artifact**	7.12 ± 1.58	7.28 ± 1.18	<0.05
**Patient Positioning**	6.08 ± 2.05	6.71 ± 1.84	<0.001
**Overall Diagnostic Value for Pathology Identification**	5.59 ± 1.97	6.32 ± 1.83	<0.001

Figures (except *p*-Values) are expressed in mean ± standard deviation (0—no attempt to meet quality requirement; 1—extremely poor quality; 2—very poor quality; 3—poor quality; 4—just below minimally acceptable diagnostic quality; 5—meeting minimally acceptable diagnostic quality; 6—just above minimally acceptable diagnostic quality; 7—good quality; 8—very good quality; 9—excellent quality; 10—outstanding quality).

## Data Availability

The data are not publicly available due to ethical restrictions.

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
