# Peer review of "Use of Smart Glasses (Assisted Reality) for Western Australian X-ray Operators’ Continuing Professional Development: A Pilot Study"

_healthcare, 2024, doi:10.3390/healthcare12131253_

Round 1

Reviewer 1 Report

Comments and Suggestions for Authors

The paper explores the use of smart glasses for assisted reality in teleradiography among Western Australian x-ray operators (XROs). Through a six-month pilot program, the study evaluates its impact on XROs' competence and the usability of the smart glasses. A good paper to read. Some minor issues and question:

1. Clarity in Usability Evaluation: Provide clearer insights into the usability challenges faced by XROs.

2. Discussion of Limitations: Acknowledge and discuss any encountered limitations for a comprehensive understanding.

3. Recommendations: Offer suggestions for future research directions to enhance the study's impact.

4. Clarify rationale for questionnaire selection.

5. Provide additional context on participant selection.

6. Strengthen the discussion by elaborating on implications and potential strategies.

The paper contributes significantly to radiography education and professional development. Addressing the minor revisions would enhance clarity and impact.

Author Response

The paper explores the use of smart glasses for assisted reality in teleradiography among Western Australian x-ray operators (XROs). Through a six-month pilot program, the study evaluates its impact on XROs’ competence and the usability of the smart glasses. A good paper to read.

Response: Thank you for your comment.

Some minor issues and question:

  1. Clarity in Usability Evaluation: Provide clearer insights into the usability challenges faced by XROs.

Response: Thank you for your comment. To address this comment, the following sentence, “This is because extra time and effort were required to obtain consent, complete 2FA and control the smart glasses using voice with unfamiliar software interface.” has been added to the fifth paragraph of the Discussion section to provide clearer insights into the usability challenges faced by XROs.

  1. Discussion of Limitations: Acknowledge and discuss any encountered limitations for a comprehensive understanding.

Response: Thank you for your comment. An additional limitation, “Besides, as per the human research ethics requirements, patients who were unable to pro-vide consent during the intervention were excluded from our study but this situation was the same as the one of Rawle et al.’s study [3].” has been added to the last paragraph of Discussion section for addressing this comment.

  1. Recommendations: Offer suggestions for future research directions to enhance the study’s impact.

Response: Thank you for your comment. An additional future research direction, “In addition, it would be interesting to match the use of smart glasses with smartphone apps specifically tailored to various fields of medicine for understanding their potential mutual application in the future [55,56].” has been added to the end of second last paragraph of Discussion section for addressing this comment.

  1. Clarify rationale for questionnaire selection.

Response: Thank you for your comment. The following sentence, “Andersson et al.’s [22,23] and Yoon et al.’s [10] questionnaires were selected for developing our evaluation questionnaires because they were designed for nurses and hence these matched our XROs’ characteristics which were predominantly nurses.” has been added to Section 2.3. Evaluation of Assisted Reality Support Program for addressing this comment.

  1. Provide additional context on participant selection.

Response: Thank you for your comment. The third sentence of the first paragraph of the Materials and Methods section has been changed to “Two WACHS remote clinical centers with relatively large numbers of radiography cases performed by their XROs per year and their (two) corresponding supervising sites (expert centers) were selected for piloting the assisted reality support for six months (1 October 2023-30 March 2024). These centers were chosen because: 1. WACHS was the major employer of WA XROs; 2. approximately 1000 cases were performed by the two XROs’ centers per year in total; and 3. all four centers were far (about 1400-2500 km) away from WA capital, Perth [1].” for addressing this comment.

  1. Strengthen the discussion by elaborating on implications and potential strategies.

Response: Thank you for your comment. To address this comment, an additional future research direction, “In addition, it would be interesting to match the use of smart glasses with smartphone apps specifically tailored to various fields of medicine for understanding their potential mutual application in the future [55,56].” has been added to the end of second last paragraph of Discussion section. Besides, an extra limitation, “Besides, as per the human research ethics requirements, patients who were unable to provide consent during the intervention were excluded from our study but this situation was the same as the one of Rawle et al.’s study [3].” has been added to the last paragraph of Discussion section.

The paper contributes significantly to radiography education and professional development. Addressing the minor revisions would enhance clarity and impact.

Response: Thank you for your comment. Revisions for addressing your comments have been made. We hope you will find our revisions satisfactory.

Reviewer 2 Report

Comments and Suggestions for Authors

Dear Authors,

I have read the manuscript with interest and some questions raised. Enlisted please find my comments.

Overall comments

The main question addressed by the research is to implement a six-month pilot program for Western Australian x-ray operators (XROs) to use smart glasses for obtaining assisted  reality support in their radiography practice from their supervising radiographers, and evaluate its  effectiveness in terms of XROs’ competence improvement and equipment usability.

The text, compared with other recently published research, add new perspectives to the last research perspectives in the specific topic area.

The information given in the text is relevant original and up to date. The research addresses a specific gap in the current research in field of radiology

Introduction, Methodology and results could be improved following Specific improvements reported in the section below.

The final considerations address the main question posed. Future perspectives are quite important.

References can be refined and updated.

Specific comments

Overall. General English grammar revision (Minor spelling errors).

Key words. “radiology” could be added in my opinion.

Introduction. Authors stated “This radiographer’s unavailability can be attributed to reasons, e.g. challenge associated with full-time radiographer recruitment, radiographer’s position deeming redundant as a result of small rural population, etc”. Please add a reference for this statement.

Materials and Methods. Authors stated “centers with relatively large numbers of radiography cases performed by their XROs per  year (about 1000 cases in total) and their (two) corresponding supervising sites (expert centers) were selected for piloting the assisted reality support for six months (1 October 2023-30 March 2024).”. Please add if and how sample size calculation has been performed.Materials and Methods. Authors stated “the guidance through the Microsoft Teams (Redmond, Washington, USA) which was the only VC platform approved by WACHS for clinical use.”. Please add details about software used, version, Manufacturer, City and State.

Materials and Methods. Authors stated “Each supervising radiographer’s site was given one iPad and every XROs’ center was provided one smart glasses device. Both iPad and smart glasses were connected to their centers’ network through WiFi”. For each machinery used, please add details about commercial name manufacturer, City and State.

Materials and Methods. Please add a reference for each method.

Materials and Methods. For each material used, please add details about commercial name manufacturer, City and State.

Statistics. Authors stated “paired t-test was used for comparing the mean  values of the pre- and post-intervention pairs of the 11-point scale items to determine any  XROs’ radiography competence (including image quality) improvements”. Was data tested for normality before paired t test application?

Results. P values have been reported in the tables but not in the main text. Please add P values all along this section.

Discussion. Authors stated “Also, smart phones with high mobility  can be given to supervising radiographers for providing the assisted reality support any- where and anytime [51].”. Provide a general interpretation of the results in the context of other evidence, and implications for future research. It could be added that “Additionally it would be interesting in the future to match the use of Smart Glasses in combination with smartphone applications specifically tailored in various fields of medicine (Sánchez-Rodríguez MT, Pinzón-Bernal MY, Jiménez-Antona C, Laguarta-Val S, Sánchez-Herrera-Baeza P, Fernández-González P, et al. Designing an Informative App for Neurorehabilitation: A Feasibility and Satisfaction Study by Physiotherapists. Healthcare (Basel). 2023 Sep 14;11(18):2549. doi: 10.3390/healthcare11182549. PMID: 37761746; PMCID: PMC10530788.) and dentistry (Pascadopoli M, Zampetti P, Nardi MG, Pellegrini M, et al. Smartphone Applications in Dentistry: A Scoping Review. Dent J (Basel). 2023 Oct 20;11(10):243. doi: 10.3390/dj11100243.) in order to understand their potential mutual application”. These concerns should be added to Discussion section.

Figure 1 a: quite not useful. It could be cropped

Comments on the Quality of English Language

General English grammar revision (Minor spelling errors).

Reviewer 3 Report

Comments and Suggestions for Authors

Dear authors of the manuscript entitled ‘Use of Smart Glasses (Assisted Reality) in the Continuing Professional Development of X-ray Operators in Western Australia: A Pilot Study’. The study was designed and conducted correctly. The introduction of the article provided an excellent introduction to the topic of the study. 

For the results, I have only two suggestions according to the tables:

Table 4. The table heading should also be on the next page when the table is split.

Table 5. From my point of view, the first-row ‘ Examination’ should be presented differently, as it is difficult to read.

The discussion section obtains other important studies in this area and compares the results of the authors and different sizes of patient groups. As an added value, it is worth mentioning that the disadvantages of smart glasses are also presented.

Author Response

Dear authors of the manuscript entitled ‘Use of Smart Glasses (Assisted Reality) in the Continuing Professional Development of X-ray Operators in Western Australia: A Pilot Study’. The study was designed and conducted correctly. The introduction of the article provided an excellent introduction to the topic of the study.

Response: Thank you for your comment.

For the results, I have only two suggestions according to the tables:

Table 4. The table heading should also be on the next page when the table is split.

Response: Thank you for your comment. The table heading, “Table 4. Participants’ perceptions of assisted reality support and equipment usability before and after intervention. Cont.” has been added to the second page of Table 4 for addressing this comment.

Table 5. From my point of view, the first-row ‘ Examination’ should be presented differently, as it is difficult to read.

Response: Thank you for your comment. The presentation style of Table 5 has been changed for improving the readability to address this comment.

The discussion section obtains other important studies in this area and compares the results of the authors and different sizes of patient groups. As an added value, it is worth mentioning that the disadvantages of smart glasses are also presented.

Response: Thank you for your comment.

Reviewer 4 Report

Comments and Suggestions for Authors

In this study, the authors investigate the usage of smart glasses to provide remote support to Western Australian X-ray operators (XROs) and assess their usefulness in increasing radiography capabilities and equipment use.

The manuscript is written on a very current topic, and the writing, the methods, as well as the discussion section, is quite good. The results are clear and understandable. The authors stated "no study has investigated its use in teleradiography". In this respect, it will be the first in the literature. The most important limitation of the article: as the authors state, "Only three XROs and two supervising radiographers completed the post-intervention questionnaires."

However, I have a few suggestions:

1. While the term "assisted reality" is used in the title of the manuscript, the term "augmented reality" appears in the keywords. These terms need to be reviewed.

2. If possible, can numbers of XROs and supervising radiographers and numbers of examinations and images be written in the abstract section to show the population of the study?

3. Only "Refusal to consent to participation / unable to obtain consent" was written in the exclusion criteria. Couldn't any factor related to the patient have affected the quality of the X-ray images? For example, loss of consciousness or the patient being in extreme pain. If factors that could affect the patient's image quality were not considered, should you add this as a limitation? or what is your opinion on the potential for patient factors to affect the image quality?

4. The tables are generally well-prepared, but I suggest that the authors review them once more to see if they can be presented in a simpler manner. 

5. There is distortion in the data format in some tables.

6. I think, the noun "x-ray" is often capitalized as "X-Ray". However, the final decision must of course be made by the authors.

Except from these minor suggestions, I believe the manuscript is excellent, will make a significant contribution to the literature, and is a high-quality study.

Author Response

In this study, the authors investigate the usage of smart glasses to provide remote support to Western Australian X-ray operators (XROs) and assess their usefulness in increasing radiography capabilities and equipment use.

The manuscript is written on a very current topic, and the writing, the methods, as well as the discussion section, is quite good. The results are clear and understandable. The authors stated "no study has investigated its use in teleradiography". In this respect, it will be the first in the literature. The most important limitation of the article: as the authors state, "Only three XROs and two supervising radiographers completed the post-intervention questionnaires."

Response: Thank you for your comment.

However, I have a few suggestions:

  1. While the term "assisted reality" is used in the title of the manuscript, the term "augmented reality" appears in the keywords. These terms need to be reviewed.

Response: Thank you for your comment. The words, “augmented reality” have been removed from the Keywords section for addressing this comment.

  1. If possible, can numbers of XROs and supervising radiographers and numbers of examinations and images be written in the abstract section to show the population of the study?

Response: Thank you for your comment. The final sentence of the Abstract has been changed to “Our study’s findings based on 13 participants (11 XROs and 2 supervising radiographers) and 2053 X-Ray images show that the assisted reality support helped to improve the XROs’ radiography competence (specifically x-ray image quality) with mean post-intervention competence values, 6.16-7.39 (out of 10) and statistical significances (p<0.001-0.05) and the equipment was considered effective for this purpose but not easy to use.” for addressing this comment.

  1. Only "Refusal to consent to participation / unable to obtain consent" was written in the exclusion criteria. Couldn't any factor related to the patient have affected the quality of the X-ray images? For example, loss of consciousness or the patient being in extreme pain. If factors that could affect the patient's image quality were not considered, should you add this as a limitation? or what is your opinion on the potential for patient factors to affect the image quality?

Response: Thank you for your comment. The following limitation, “Besides, as per the human research ethics requirements, patients who were unable to pro-vide consent during the intervention were excluded from our study but this situation was the same as the one of Rawle et al.’s study [3].” has been added to the last paragraph of Discussion section for addressing this comment.

  1. The tables are generally well-prepared, but I suggest that the authors review them once more to see if they can be presented in a simpler manner.

Response: Thank you for your comment. To address this comment, the following changes have been made to Tables 4 and 5.

The table heading, “Table 4. Participants’ perceptions of assisted reality support and equipment usability before and after intervention. Cont.” has been added to the second page of Table 4.

The presentation style of Table 5 has been changed for improving the readability.

  1. There is distortion in the data format in some tables.

Response: Thank you for your comment. The tables have been reviewed and data format changes have been made to Tables 1 and 4 for consistency and addressing this comment.

  1. I think, the noun "x-ray" is often capitalized as "X-Ray". However, the final decision must of course be made by the authors.

Response: Thank you for your comment. The word, “x-ray” has been changed to “X-Ray” for addressing this comment.

Except from these minor suggestions, I believe the manuscript is excellent, will make a significant contribution to the literature, and is a high-quality study.

Response: Thank you for your comment.

Round 2

Reviewer 2 Report

Comments and Suggestions for Authors

Dear Authors,

The manuscript has been revised taking into careful account all the comments.

Introduction is complete.

Materials and Methods are correct.

Results are clear.

Discussion is wide and well written.

Conclusions are sound.

References are adequate.

Figures and Tables are readable and explanatory.

Thank you.